

# Salt marsh sediment bacterial communities maintain original population structure after transplantation across a latitudinal gradient

Angus Angermeyer[1,2,3], Sarah C. Crosby[1,2,4] and Julie A. Huber[2,5]

[1] Ecology and Evolutionary Biology, Brown University, Providence, RI, USA
[2] Josephine Bay Paul Center, Marine Biological Laboratory, Woods Hole, MA, USA
[3] Plant and Microbial Biology, University of California, Berkeley, Berkeley, CA, USA
[4] Harbor Watch, Earthplace Inc., Westport, CT, USA
[5] Marine Chemistry and Geochemistry, Woods Hole Oceanographic Institution, Woods Hole, MA, USA

Corresponding author
Angus Angermeyer,
angus.angermeyer@gmail.com

## ABSTRACT

Dispersal and environmental selection are two of the most important factors that govern the distributions of microbial communities in nature. While dispersal rates are often inferred by measuring the degree to which community similarity diminishes with increasing geographic distance, determining the extent to which environmental selection impacts the distribution of microbes is more complex. To address this knowledge gap, we performed a large reciprocal transplant experiment to simulate the dispersal of US East Coast salt marsh *Spartina alterniflora* rhizome-associated microbial sediment communities across a latitudinal gradient and determined if any shifts in microbial community composition occurred as a result of the transplantation. Using bacterial 16S rRNA gene sequencing, we did not observe large-scale changes in community composition over a five-month *S. alterniflora* summer growing season and found that transplanted communities more closely resembled their origin sites than their destination sites. Furthermore, transplanted communities grouped predominantly by region, with two sites from the north and three sites to the south hosting distinct bacterial taxa, suggesting that sediment communities transplanted from north to south tended to retain their northern microbial distributions, and south to north maintained a southern distribution. A small number of potential indicator 16S rRNA gene sequences had distributions that were strongly correlated to both temperature and nitrogen, indicating that some organisms are more sensitive to environmental factors than others. These results provide new insight into the microbial biogeography of salt marsh sediments and suggest that established bacterial communities in frequently-inundated environments may be both highly resistant to invasion and resilient to some environmental shifts. However, the extent to which environmental selection impacts these communities is taxon specific and variable, highlighting the complex interplay between dispersal and environmental selection for microbial communities in nature.

## INTRODUCTION

It is increasingly clear that microbial taxa vary in their ability to disperse across landscapes (*Martiny et al., 2006*), and that environmental conditions as well as biological mechanisms can influence community dynamics post-dispersal. Investigating the interplay between these variables is key to building better models of microbial biogeography. A wide variety of studies, spanning coastal salt marshes, oceans, and forest rainwater deposits often indicate a taxa–area relationship for bacteria, with both geographic distance and environmental factors giving rise to uneven spatial distributions of microbes in nature (*Angermeyer, Crosby & Huber, 2015*; *Horner-Devine et al., 2004*; *Zinger, Boetius & Ramette, 2014*; *Bell et al., 2005*). These distance–decay relationships are thought to be driven by the interplay between the mechanisms of dispersal, environmental selection, genetic drift, and random mutation (*Hanson et al., 2012*; *Nemergut et al., 2013*). However, the details of how each mechanism or combination of mechanisms affects the composition of complex natural communities is not well understood. For instance, both selection and drift act upon microorganisms after a dispersal event, but the rates of those events are likely to be highly variable across the diversity of taxa that comprise a complex microbial community (*van der Gast, 2014*). This variability can be caused by neutral effects such as abundance (higher abundance taxa have a better chance of dispersing) (*Livermore & Jones, 2015*) or due to active dispersal-mechanism phenotypes (such as biofilm attachment/detachment) (*McDougald et al., 2011*). Taxa that are in low abundance can be more strongly influenced by bottleneck effects in comparison to more abundant, highly active groups. A further challenge in studying the mechanisms that drive distance–decay patterns is in the detection of subtle genetic flows caused by drift and mutation in a diverse natural community (*Andam et al., 2016*).

Here, we build upon our previous work on microbial biogeography in a salt marsh model system (*Angermeyer, Crosby & Huber, 2015*) to examine dispersal and adaptation of the resident bacterial communities in more detail using a reciprocal transplant approach. Reciprocal transplantation is a well-established tool in traditional "macro" ecology, but is infrequently utilized in microbial biogeography. Reciprocal transplantation effectively simulates a large dispersal event and affords the opportunity to then observe the effects of such an event on microbial communities (*Balser & Firestone, 2005*; *Waldrop & Firestone, 2006*). Transplants leverage both common garden and environmental treatment techniques to test for the selection effects of one environment on multiple communities as well as the effects of multiple environmental gradients on the replicates of each community (*Reed & Martiny, 2007*). However, transplant experiments generally cannot directly observe the rates of genetic drift and mutation due to the timescales over which these phenomena likely manifest, as well as the high frequency and depth of DNA sequencing that would be required to detect them.

Previous studies explored how the functional parameters of microbial communities can shift (*Gasol et al., 2002*; *Comte, Fauteux & del Giorgio, 2013*) and how community taxonomic composition can change (*Oakley et al., 2010*; *Bell, 2010*) after transplantation. Microbial reciprocal transplants have been performed across a wide range of

environmental systems including: alpine soils (*Rui et al., 2015*), wetland sediments (*Reed & Martiny, 2013*; *Morrissey & Franklin, 2015*), and decomposing plant litter (*St John, Orwin & Dickie, 2011*), among others. These studies employed diverse methodological approaches to reciprocal transplantation which includes sealing samples to prevent invasion from host site communities (with nutrient permeability) (*Bell, 2010*; *Reed & Martiny, 2013*; *Morrissey & Franklin, 2015*), inverting sea-ice cores to alter the photosynthetic regime vertically (*Martin et al., 2011*), and swapping gut microbiota between gnotobiotic zebra fish and mice (*Rawls et al., 2006*). The duration of the transplant (time between transplantation and sample recovery) experiments also varies widely, with studies ranging from two weeks (*Reed & Martiny, 2013*) to 17 years (*Bond-Lamberty et al., 2016*). While a relatively low number of microbial reciprocal transplant studies have been performed, each study lends important insight into the ecology and biogeography of microbial communities of a specific environment.

In this study, we considered transplants as simulated dispersal events and asked the question: Do microbial communities undergo compositional changes in response to mass dispersal events between similar environments (salt marshes) across a latitudinal gradient? If so, do these responses correlate to temperature, other environmental shifts, and/or the geographic transplant-distance experienced by the dispersed community? We considered four hypotheses for changes that could occur between transplanted communities in a new host site versus control communities that remained in their original marsh site: (1) No change—The transplanted community will appear identical to communities from its original site. This is the null hypothesis indicating no invasion and no environmental selection occurred in the transplanted community; (2) total adoption—The transplanted community will change to appear identical to the communities in its new host marsh. This result indicates that either the transplanted sediments were completely invaded and replaced by the surrounding marsh communities, or that environmental selection is overwhelmingly able to reshape the community structure, or a combination of both; (3) random shift—The transplanted community will change to something distinct from both origin and host communities. In this case, the cause could be either a fundamental problem with the transplantation methodology (e.g., contamination) or the effects of unknown and unmeasured environmental variables; and (4) host shift—The transplanted community will change to something in between origin and host communities. This result indicates that invasion and/or environmental selection is affecting the transplanted community, but it is either recalcitrant to total adoption or the process takes longer than the duration of the experiment.

Here we present results of this reciprocal transplant experiment of *Spartina alterniflora* rhizome-associated bacterial communities between five salt marshes along the US East Coast to examine the roles that dispersal and environmental selection play in driving microbial community structure. The approach builds on previous work of biogeography of microbial communities in salt marsh sediments and provides new insights into the impacts of microbial dispersal and selection in the environment.

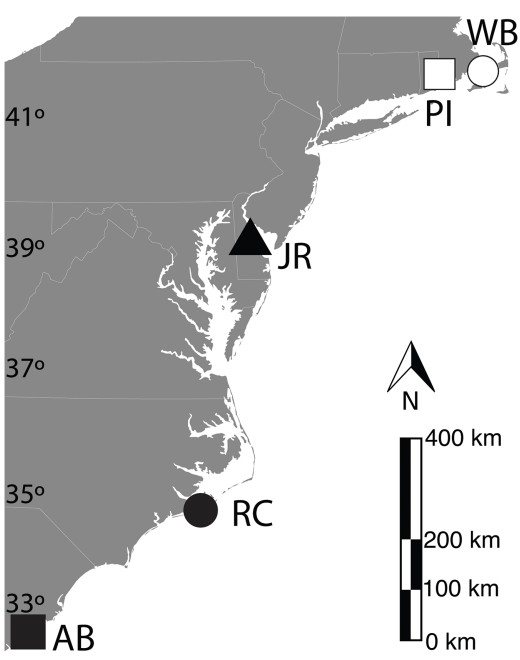

**Figure 1 Geographic locations of U.S. easy coast study sites.** Locations of salt marsh sediment sampling sites. WB-Waquoit Bay, PI-Prudence Island, JR-Jones River, RC-Rachael Carson, AB-ACE Basin. Symbols are used to refer to each site in following figures. Base map provided by https://freevectormaps.com/.

## MATERIALS AND METHODS

### Reciprocal transplantation

Fifteen sediment cores with associated live *S. alterniflora* roots and stems were collected from each of five US East Coast salt marshes in March 2010 (15/marsh; 75 total): Waquoit Bay National Estuarine Research Reserve (WB)—East Falmouth, MA [41.5806N, 70.5198W]; Prudence Island National Estuarine Research Reserve (PI)—Portsmouth, RI [41.6249N, 71.3228W]; St. Jones River National Estuarine Research Reserve (JR)—Dover, DE [39.0889N, 75.4363W]; Rachel Carson National Estuarine Research Reserve (RC)—Beaufort, NC [34.7221N, 76.6798W]; ACE Basin National Estuarine Research Reserve (AB)—Bennett's Point, SC [32.5576N, 80.4365W] (Fig. 1). All cores were collected from within monospecific stands of tall-form clonal *S. alterniflora* with similar peak stem heights that were collected at least 1 m away from the nearest creek-bank and in a region of each marsh that experienced semi-diurnal tides with less than 20% submergence per day (*Crosby et al., 2015*, *2017*).

Sediment cores measured 8 cm in diameter by ~25 cm long and were extracted from the marsh using serrated stainless-steel manual corer, taking care not to disturb the internal structure. Triplicate cores from each marsh site were transplanted in a fully factorial manner to every other marsh site, including a control set re-planted into the origin site. For example, from the Waquoit Bay salt marsh site, three cores were transplanted to Prudence Island, three to St. Jones River, three to Rachel Carson, three to ACE Basin, and three were replanted back into Waquoit Bay. The same process was

repeated for 15 cores from each other site. All transplanted and replanted cores were maintained at approximately 20 °C (air-conditioned during day, cool ambient at night) and covered with damp cloth during transport to retain moisture. Cores were kept in these conditions for one to seven days before replanting, depending on the travel distance between sites. Cores were planted by creating a fresh core-hole in the host marsh at the appropriate depth, and carefully placing one transplanted or replanted core in each hole. Due to local site restrictions, a thin layer of high-permeability, "weed-blocking" ground cloth was placed around all cores transplanted into the St. Jones River marsh.

All 75 cores were recovered from each marsh in October 2010, a six-month transplant duration. Approximately 2 g of sediment were collected with a sterilized scoopula from an undisturbed region 20 cm below the surface and in the center of each core. Each sample was suspended in LifeGuard (Mo Bio, Carlsbad, CA, USA) to preserve its nucleic acids. Sediment was collected in an identical fashion from triplicate, newly extricated "pristine" cores at each site. The total number of sediment samples was therefore 18 per marsh (three from each of the other four sites (12 total), three from that sites replanted controls, and three pristine new samples from the site). Two transplant samples were unable to be recovered (PI-to-JR #2 and WB-to-AB #3), resulting in a total of 88 samples.

## Environmental measurements

Three sediment samples per site (~5 g each) were collected in October 2010 when the transplants were recovered and immediately frozen on dry ice. These samples were pooled by site, dried at 70 °C for four days, and were analyzed according to *Meyer et al. (2013)* using a Thermo Scientific CN Analyzer (Model Flash 2000; ThermoFisher Scientific, Wilmington, DE, USA) to determine concentrations of total carbon and nitrogen. Mean air temperature over the course of the transplant experiment was collected from the nearest National Oceanic and Atmospheric Administration weather station ([Site: Station ID] WB: WAXM3, PI: PTCR1, JR: DRSD1, RC: BFTN7, AB: ACXS1). Mean average salinity was determined from measurements taken in March, May, July, and October at each site over the same time period using a YSI Model 85 water quality meter (YSI, Yellow Springs, OH, USA) (*Crosby et al., 2017*). Pairwise Mantel tests (*Mantel, 1967*) and linear regressions were performed between all measured environmental variables to identify possible correlations.

## DNA extraction

Nucleic acids from each sediment sample were extracted using the MoBio Powersoil RNA/DNA extraction kits (MoBio, Carlsbad, CA, USA), eluted in the provided buffer, and quantified on a Nanodrop 2000 (ThermoFisher Scientific, Wilmington, DE, USA). Extracted DNA samples were diluted to achieve ~10–15 ng/µl standardized concentrations for downstream applications. DNA was stored at −80 °C.

### 16S rRNA gene PCR, sequencing and analysis

Bacterial amplicon generation and sequencing was performed as described previously (*Angermeyer, Crosby & Huber, 2015*). Briefly, the v4v5 regions of the bacterial 16S rRNA gene were amplified and prepared according to *Huse et al. (2014)*. Sequencing was performed on an Illumina MiSeq sequencer (Illumina, San Diego, CA, USA) at the W. M. Keck Ecological and Evolutionary Genetics Facility at the Marine Biological Laboratory in Woods Hole, MA, USA. Reads were merged and quality checked using illumina-utils (*Eren et al., 2013a*, *2013b*) (code available at https://github.com/meren/illumina-utils) using a modification of earlier methods (*Huse et al., 2007*). Sequence data was subsampled (rarefication) to ensure even sampling across all samples. Operational taxonomic unit (OTU) clustering at 97% similarity and taxonomic assignments were performed using QIIME (pick_otus.py, method: uclust) (*Caporaso et al., 2010a*, *2010b*) and GAST (*Huse et al., 2008*). Sequences are publicly available at the "The Visualization and Analysis of Microbial Population Structures" (VAMPS) website (https://vamps.mbl.edu/) (*Huse et al., 2014*) under project title: "JAH_TRP_Bv4v5" and are deposited at the NCBI SRA under accession number PRJNA384656.

### Environmental correlations of controls

16S rRNA gene OTU data for replanted controls and pristine controls were extracted from the complete data set across all sites and used to construct Bray–Curtis dissimilarity matrices in mothur (*Schloss et al., 2009*). This was completed for all OTUs as well as for the top 10 most abundant OTUs across all samples, which produced four distinct matrices (Replant_all, Replant_top, Pristine_all, and Pristine_top). Pairwise linear regressions were generated in R (*R Core Team, 2014*) using the "`pairs()`" function between each control matrix and geographic distance (to minimize skew, all dissimilarity matrices and GeoDist were natural log transformed for this analysis (*Martiny et al., 2011*). A pairwise Mantel test was performed concurrently with 999 permutations.

### Ordination analyses

The 16S rRNA gene OTU data were parsed into two subsets: all OTUs (the entire dataset), and top 10 OTUs (the 10 most abundant OTUs across all sites and samples). For each ordination plot, the specific samples of interest were extracted from a subset (e.g., only pristine and transplant control samples). Non-metric multidimensional scaling (nMDS) calculations were performed using the "`nmds()`" function in mothur with 999 iterations.

### Potential Indicator OTUs

The observed abundances of each specific OTU across all samples were compared to the degree of environmental change and geographic distance measured between a transplant sample's origin site and destination site. For each sample, this provided an ($x$, $y$) coordinate data-point which, when taken together, created a multi-coordinate dataset. To have spatial consistency between datasets, it was necessary to calculate percent OTU abundance per sample (observed OTU abundance in a sample/total abundance of

**Table 1 Site locations and environmental variables.**

| Site | Lat. (°N) | Lon. (°W) | %N (Wt) | %C (Wt) | Salinity (ppt) | Air temp (°C) |
|------|-----------|-----------|---------|---------|----------------|---------------|
| WB | 41.580 | 70.521 | 0.92 | 13.85 | 25.1 | 17.1 ± 6.3 |
| PI | 41.625 | 71.324 | 1.34 | 23.58 | 26.6 | 17.8 ± 6.5 |
| JR | 39.089 | 75.437 | 0.52 | 5.99 | 10.7 | 20.1 ± 7 |
| RC | 34.723 | 76.675 | 0.57 | 8.78 | 32.3 | 23 ± 5.6 |
| AB | 32.558 | 80.439 | 0.51 | 8.59 | 26.4 | 24.2 ± 5.7 |

**Note:**
Sites are listed north to south. Nitrogen and carbon values are percent by weight.

that OTU across all samples). Similarly, environmental data was natural log transformed to facilitate comparison between variables. For example, if an OTU abundance was 100 in transplant sample "WB-to-AB" and that OTU total abundance was 1,000 across all samples, the percent OTU abundance per sample would be 10%. If temperature was the variable being considered, the ln-transformed temperature increase from site WB-to-AB is: $\ln[24.2\,°C]-\ln[17.1\,°C] = 0.296$ (Table 1). The $(x, y)$ coordinate would therefore be (0.296, 0.10) with abundance as the dependent variable. If the next sample to be evaluated were AB-to-WB (the opposite direction) and the OTU abundance in transplant sample "AB-to-WB" was 10 (10/1000 = 0.01), then the coordinate would be (−0.296, 0.01). For each OTU and for each environmental variable (excluding percent carbon), an $x$-$y$ dataset was populated in this fashion. The results were also curated to only include OTUs that occurred in >95% of the samples (at least 84 of 88 samples).

A least-squares linear regression test was then performed on each dataset using the "`scipy.stats.linregress()`" function in the SciPy Python module (*Jones, Oliphant & Peterson, 2001*) to calculate the slope and $R$-squared value of the best linear fit. To minimize the possibility of type I error (false positives) when searching for correlations in a large dataset (63337 OTUs × 4 variables = 253,348 regressions), a Bonferroni correction was applied to an initially conservative significance cutoff ($\alpha = 0.001$) resulting in a per-hypothesis corrected-confidence level of $\alpha = 3.9e^{-9}$ ($\alpha$ /# regression tests). Therefore, a regression's correlation was rejected if a two-sided hypothesis test (null: slope = 0) returned a $P$-value $\geq 3.9e^{-9}$, which approximately corresponds to $R$-squared values >0.40.

# RESULTS

## Transplant recovery

Upon completion of the transplant experiment, the cores were still distinct from the surrounding host marsh sediments, with a visibly clear delineation between core and core hole and little-to-no root intrusion between core and surrounding sediment. Most were recovered with only minor re-coring. However, there was some variability in the structure of the recovered cores. The cores originating from Waquoit Bay, Prudence Island, and Jones River (the northern three sites) were almost entirely intact regardless of the destination host site. Cores from Rachel Carson and especially Ace Basin suffered some loss of sediment due presumably to tidal erosion. Northern sediments have more peat and greater root structure whereas southern sites are muddier and tend to

have fewer, larger roots (*Crosby et al., 2015*). Nevertheless, in all cases there was ample core remaining to recover the necessary amount of sediment from 20 cm below the surface from the uncontaminated center of the core. It was also observed that the survival of the *Spartina* plants themselves varied by origin. The further south a plant was transplanted, the more likely the stems were to die, however plants originating from the southern sites showed little to no mortality regardless of how far north they were transplanted (*Crosby et al., 2017*).

## Environmental characteristics

Variability between sites was similar to observations made at the same sites during other parts of the season (*Angermeyer, Crosby & Huber, 2015*). Air temperature ranged from 17.1 to 24.2 °C, as expected along the latitudinal gradient. Salinity was lowest at JR (10.7 ppt) due to its greater distance from the ocean. The other sites were closer to the coast and had higher salinity (25.1–32.3 ppt). Percent total nitrogen by weight (0.51–1.34%) and carbon by weight (8.59–23.58%) were lowest in the southern three sites and relatively higher in the north with a sharp increase at PI (Table 1). Pairwise mantel tests ($M$) and linear regression ($R^2$) confirmed that temperature and geographic distance were very strongly correlated ($M = 0.98$, $R^2 = 0.96$, $P \leq 0.05$) as well as nitrogen and carbon ($M = 0.97$, $R^2 = 0.94$, $P \leq 0.05$, Fig. S1). There were also significant correlations between both carbon/nitrogen and geodist/temperature pairs ($M = 0.35$–$0.49$, $R^2 = 0.12$–$0.24$, $P \leq 0.05$, Fig. S1).

## Bacterial community composition

The average concentration of extracted DNA was 49.12 ng/µl (stdev ± 24.47) before standardization. Sequencing of the v4v5 region of the bacterial 16S rRNA gene generated 3,434,579 high quality sequences with an average length of 376 base pairs. There were an average of 39,029 (SD ± 11,212) sequences in each sample before rarefying to an even depth of 14,376 sequences per sample. After rarefaction, taxonomic identification revealed 57,667 total OTUs across samples at 97% similarity. Of these, 17,154 singletons were removed, leaving 40,513 OTUs for subsequent analyses.

   Analyses of the 16S rRNA gene data revealed a clear difference in bacterial community composition between the sediment samples that originated in the northern sites (WB, PI) versus the three southern sites (JR, RC, AB) (Fig. 2). When a sample originated in the south, its bacterial community tended to be dominated by *Gammaproteobacteria* (average relative abundance across all samples from each site: JR-50%, RC-43%, AB-33%). Within this class, one OTU belonging to the genus *Vibrio* (OTU# 32074) dominated in these southern samples (average relative abundance across all samples from each site: JR-20%, RC-23%, AB-23%). In contrast, *Gammaproteobacteria* were at much lower relative abundances (7–8%) in the two northern sites, and the relative abundance of the same *Vibrio* OTU was also extremely low (<0.3%). However, the northern sites were more abundant in *Anaerolineae* (phylum *Chloroflexi*) (average relative abundance across all samples from each site: WB-16%, PI-12%) and *Deltaproteobacteria* (average relative abundance across all samples from each site:
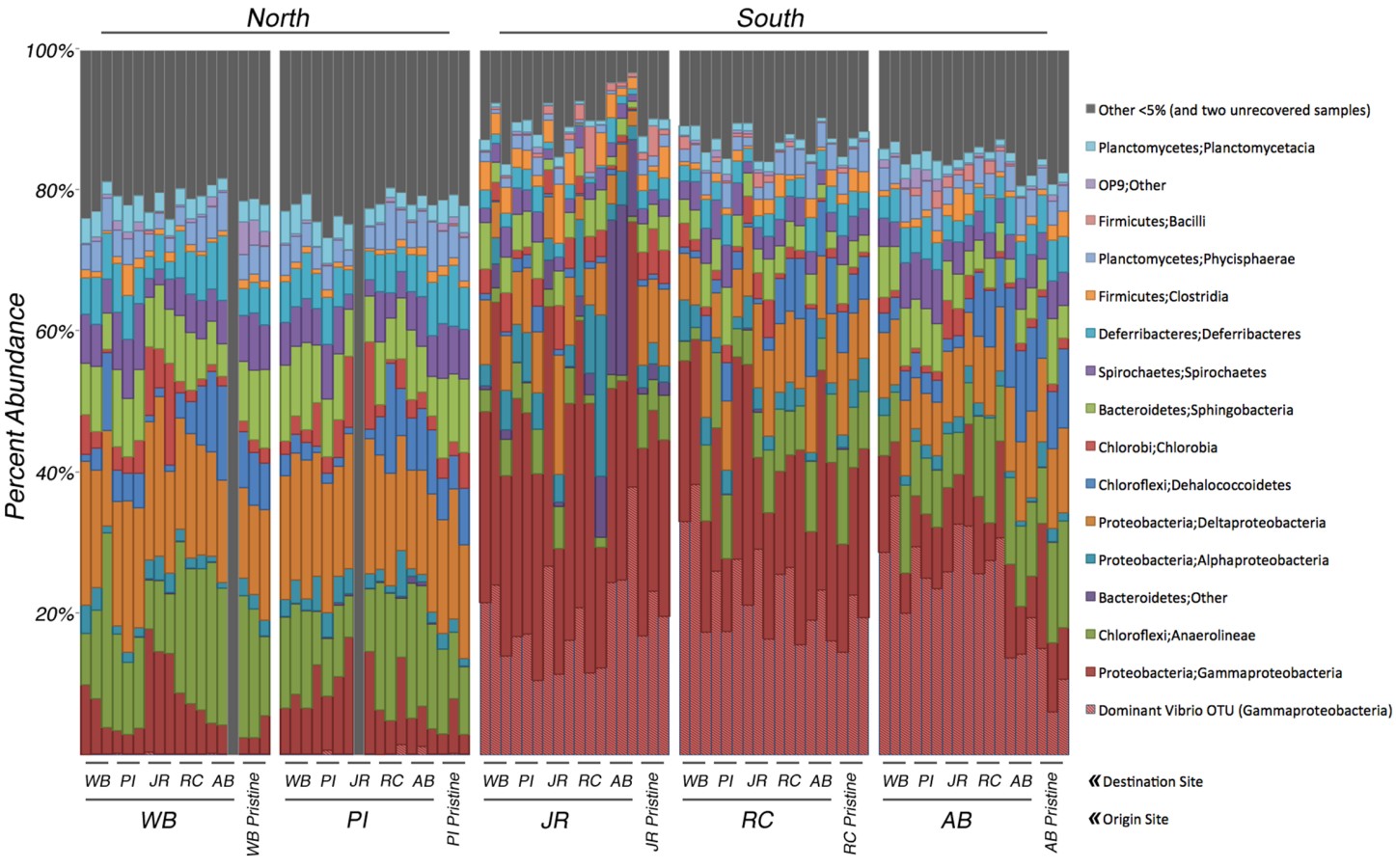

**Figure 2 Relative abundance of bacterial 16S rRNA genes among transplant samples and controls.** Taxonomic assignments for bacterial 16S rRNA genes were based on 97% sequence similarity. Only taxa that occur in ≥5% abundance in any individual sample are colored. Blocks of samples are arranged by origin site (north to south), and by destination site within each block. The last three columns in each block are the pristine controls. The hatched columns are the *Vibrio* sp. subset of Gammaproteobacteria. Grey columns represent all taxa <5% abundance and two samples that did not amplify.

WB-17%, PI-17%) compared to the southern sites, which averaged 3–6% for *Anaerolineae* and 8–10% for *Deltaproteobacteria*. These trends in relative abundance were consistent for all transplant samples and pristine samples.

## Analysis of control samples

Mantel tests and linear regressions of 16S rRNA gene OTUs determined that the transplant control (cores re-planted in the same marsh) and pristine control (new cores collected from each marsh at the end of the transplant experiment) microbial communities were correlated when comparing all OTUs (Replant_all vs. Pristine_all: $M = 0.69$, $R^2 = 0.47$, $P \leq 0.05$). These statistical tests further showed that datasets comprised of only the top 10 most abundant 16S rRNA gene OTUs across all samples were also correlated between replant and pristine controls (Replant_top vs. Pristine_top: $M = 0.71$, $R^2 = 0.50$, $P \leq 0.05$) (Fig. S2). Additionally, the pristine datasets exhibited a significant correlation with the geographic distance between sites, which is consistent with

previous findings (*Angermeyer, Crosby & Huber, 2015*) (Pristine_all vs. GeoDist: $M = 0.82$, $R^2 = 0.68$, $P \leq 0.05$) and (Pristine_top vs. GeoDist: $M = 0.75$, $R^2 = 0.56$, $P \leq 0.05$) (Fig. S2).

## Ordination analysis

Non-metric multidimensional scaling plots of the Bray–Curtis similarities for 16S rRNA gene OTUs between transplant control and pristine control samples illustrated that the two controls grouped together by site (Figs. 3A and 3B). This was true for both all OTUs and the top 10 most abundant OTUs, although the stress value was relatively high for all OTUs (Figs. 3A and 3B). When expanded to include all transplant samples, as well as both controls, labeled by origin site (Figs. 3C and 3D), the transplant communities grouped more closely with the sites they originated from, particularly when only the top 10 most abundant OTUs were considered. However, when labeled by destination site, there were no obvious grouping patterns (Fig. S3).

## Potential indicator OTUs

The linear regression analysis identified four potential correlations among three 16S rRNA gene OTUs (*Chloroflexi; Anaerolineae; Anaerolineales; Anaerolinaceae* [OTU# 21858], *Proteobacteria; Gammaproteobacteria; Vibrionales; Vibrionaceae; Vibrio* [OTU# 32074], and *Proteobacteria; Deltaproteobacteria; Desulfobacterales; Desulfobacteraceae* [OTU# 13670]) and a change with a specific environmental variable (Table S1; Fig. 4). Two correlations were with change in temperature (*Anaerolinaceae*, *Vibrio*) and two in percent nitrogen (*Anaerolinaceae*, *Desulfobacteraceae*). Due to the environmental correlation between carbon and nitrogen, carbon results were excluded from the results. The *R*-squared values of the regressions ranged from 0.41 to 0.59 and the slopes −3.53 to 4.01. The taxonomic identity of the OTUs was determined to the family level for *Anaerolinaceae* and *Desulfobacteraceae*, and to genus for the *Vibrio* OTU. The total percent abundances across all samples for each OTU were 0.075% for *Anaerolinaceae*, 0.241% for *Desulfobacteraceae-*, and 10.3% for *Vibrio*.

## DISCUSSION

Although distance–decay analyses are an excellent approach to observing the ß-diversity relationships of microbial communities separated by a range of geographic distances, they can only examine a "snapshot" of microbial distributions in time (*Martiny et al., 2011*). Performing multiple experiments over time can provide additional insight into how communities change with temporal or seasonal patterns (*Fortunato et al., 2013*), but such experiments do not consider the role that dispersal plays in microbial community distribution patterns. To address this need, we performed a reciprocal transplant experiment to simulate the dispersal of salt marsh sediment microbial communities across a latitudinal gradient and over the course of a summer growing season in an attempt to disentangle the roles that dispersal, environmental selection, and geographic distance play in driving microbial community changes. Transplant experiments can be difficult to interpret in many cases due to the challenges of controlling

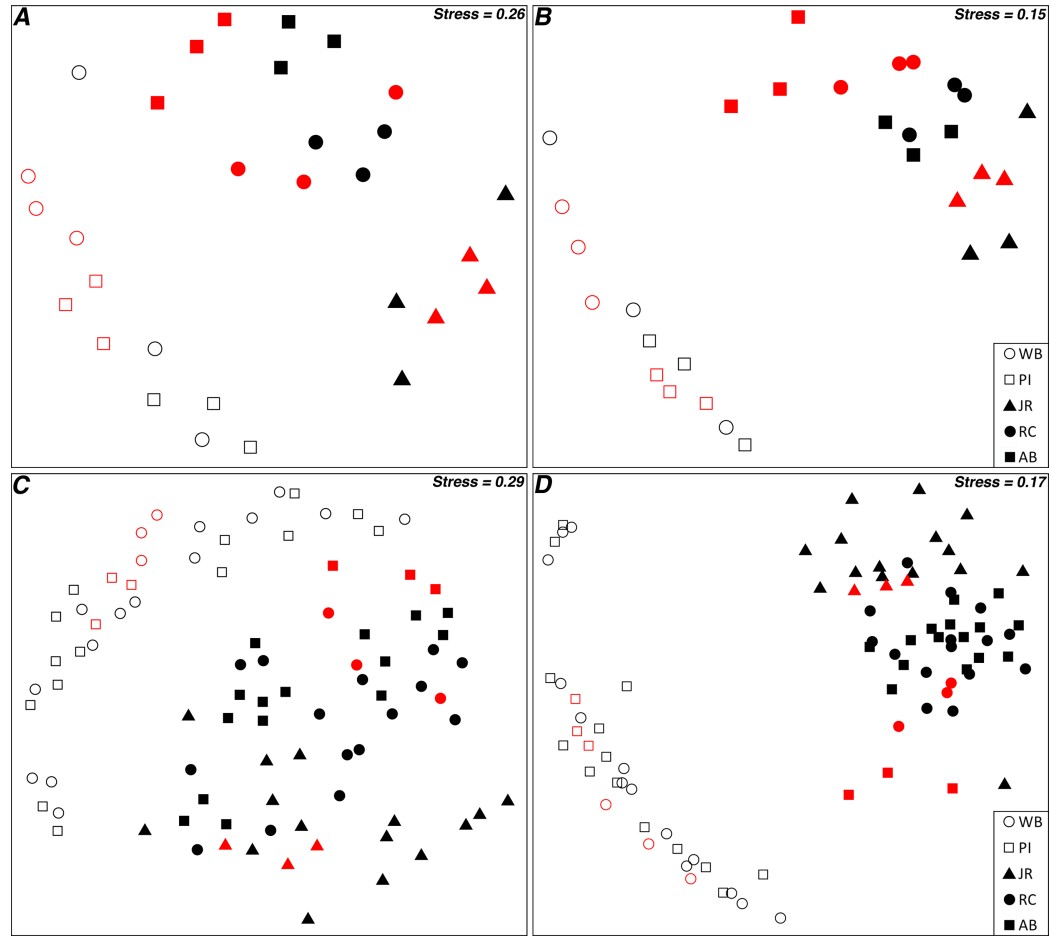

**Figure 3 nMDS ordination plots of transplant samples and controls.** (A) All 16S rRNA gene OTUs for control samples; (B) Top 10 most abundant 16S rRNA gene OTUs for control samples; (C) All 16S rRNA gene OTUs for all samples; and (D) Top 10 most abundant 16S rRNA gene OTUs for all samples. "Red" symbols are pristine control samples, "black" are transplant samples. Transplant samples are labeled by origin site.                                   

for multiple, diverse environmental variables (*Reed & Martiny, 2007*). However, the broad, ecosystem-scale environmental similarities (as compared to most other microbial transplant study systems) over a large latitudinal gradient made these marshes well-suited to a reciprocal transplantation experiment. We tested the acceptability of our core-transplant method by comparing communities from control sediments (removed from each marsh and replanted back into the same site) versus communities from undisturbed "pristine" control samples collected at the end of the experiment. An ordination analysis demonstrated that the replanted controls did not diverge greatly from the natural marsh community state and therefore the methods were likely sound (Figs. 3A and 3B). The controls used in this study also allowed us to examine the distance–decay relationships between the same sites as in our previous study (although with a smaller data set) during a different season (*Angermeyer, Crosby & Huber, 2015*), and we confirmed that the trend observed is seasonally consistent (Fig. S2).

Peerj

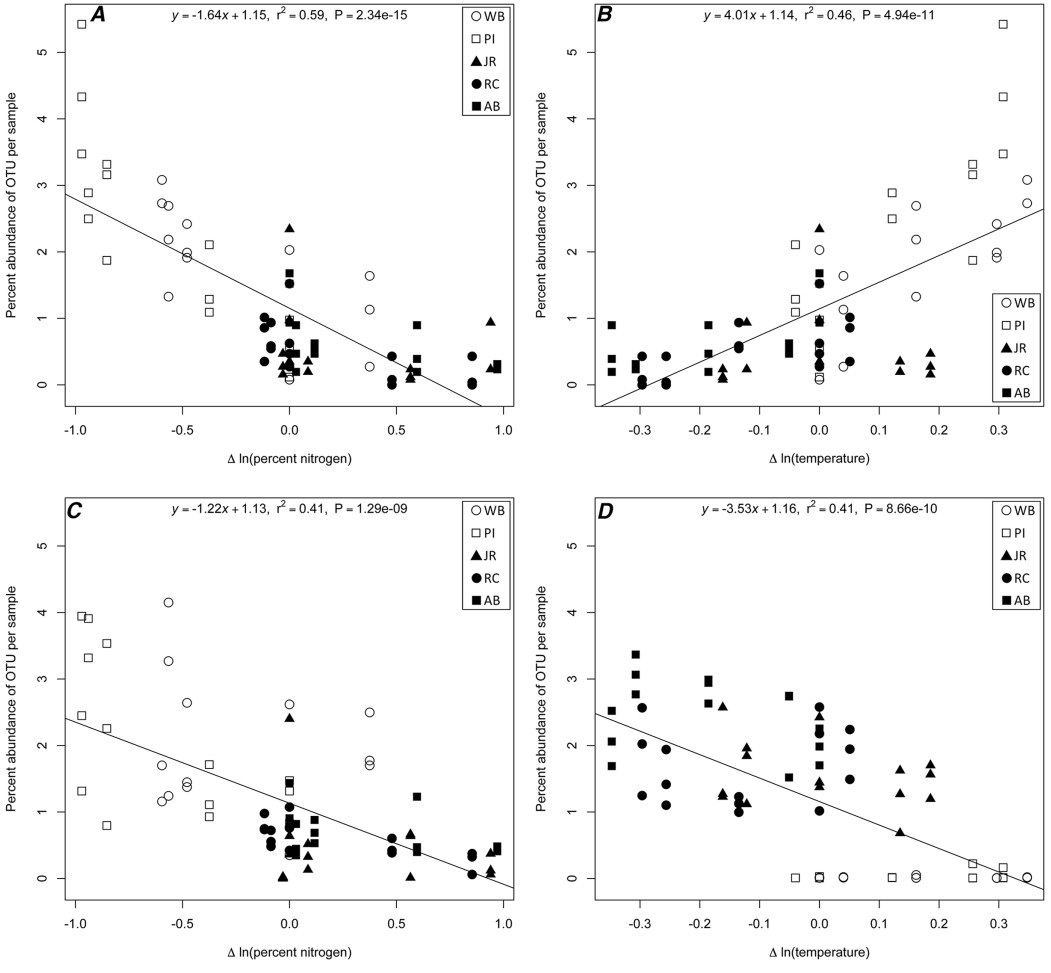

**Figure 4 Linear regression plots of indicator OTUs.** 16S rRNA gene OTUs percent abundance (by sample) as a function of change in a specific environmental variable for (A) *Anaerolinaceae* and nitrogen; (B) *Anaerolinaceae* and temperature; (C) *Desulfobacteraceae* and nitrogen; and (D) *Vibrio* and temperature. Equations, *R*-squared values and *P*-values of each linear regression are provided.

This study was designed to determine if microbial community composition changed after transplantation into a new host marsh and to what, if any, degree several measured environmental differences between origin and destination marshes influenced microbial community structure. After five months in a host marsh, transplanted communities most closely resembled the current state of the sites from which they originated (Fig. 2). Of the four hypotheses presented in the introduction (no change, total adoption, random shift, and host shift), the null hypothesis of no change best matches our results. However, ordination analysis revealed that transplanted communities grouped by latitudinal region, with Waquoit Bay and Prudence Island in the north, and Jones River, Rachael Carson, and Ace Basin in the south (Figs. 3C and 3D). In effect, this meant that sediment communities transplanted from north to south tended to retain their northern microbial distributions, whereas south to north maintained a southern distribution. This trend was especially clear when we focused on the top 10 most

abundant OTUs (Fig. 3D). It is also important to note that the two northern sites are the closest together spatially and distance–decay curves suggest they have a lower barrier to dispersal between them than between northern and southern sites in general (*Angermeyer, Crosby & Huber, 2015*).

Several recent studies have shown that transplanted communities retain compositions more closely associated with the communities at their origin sites than the host locations. In one experiment, deep-sea methane seep carbonates were reciprocally transplanted between high and low activity methane seeps (*Case et al., 2015*). The authors found evidence that microbial carbonate communities were highly resistant to invasion and maintained 49–90% of their original "high-activity site" taxa after 13 months in a low-activity site. They speculated that these taxa may be "adapted to cycles of seep quiescence and activation," which may provide ecological advantages in fluctuating environments. A second study exchanged soil samples between calcareous and siliceous glacier forefields for 15 months and found that while seasonal precipitation caused some temporary diversification of the transplanted communities, in the end, no large effect of the transplantation was observed on either soil type (*Lazzaro, Gauer & Zeyer, 2011*). In a third study, dryland soils were transplanted across an elevation gradient, and the authors found that although respiration rates of soils tended to match the host site controls, the microbial community composition retained a composition most similar to their origin site controls. This result was especially surprising given that the duration of the transplantation experiment was 17 years. It is generally hypothesized that this experiment's duration would be ample to fully acclimatize a transplanted sample to both microbial invasion and seasonal patterns (*Bond-Lamberty et al., 2016*). However, this result highlights that actual rates of drift, mutation, dispersal and selection are poorly constrained for microbes in the natural world, including how they vary between microbial lineages and habitats.

In our study, the distribution of bacterial OTUs between north and south sites revealed that the southern sites were highly enriched for one OTU belonging to the genus *Vibrio*, while the northern sites had a higher relative abundance of an *Anaerolinaceae* OTU and a *Desulfobacteraceae* OTU. We speculated that the pattern of *Vibrio* abundance, in particular, was driven by temperature differences between southern and northern sites as correlations between the abundances of various *Vibrio* taxa and temperature in late summer are well described in the mid-Atlantic US East Coast (*Pfeffer, Hite & Oliver, 2003*; *Thompson et al., 2004*). To confirm if this temperature effect was responsible for the distributions of *Vibrio* in our data and to determine if other OTUs exhibited correlations with any of the environmental variables, we performed a linear regression analysis with strict threshold cutoffs to only consider the most highly correlated OTU-environment relationships for further analysis (Table S1). Our analysis confirmed that one *Vibrio* OTU had a statistically significant correlation with the change in temperature experienced by each transplant (Fig. 4D). The negative slope indicated that as the difference in temperature from origin site to destination site increased, the *Vibrio* OTU was less abundant, meaning that in sample cores transplanted from warmer sites to cooler sites, this OTU was more abundant that

other transplanted communities. A similar pattern was observed for both *Anaerolinaceae* and *Desulfobacteraceae* OTUs (Figs. 4A and 4C) with a negative correlation to change in percent nitrogen. It was observed that in sample cores transplanted from sites high in nitrogen to sites with lower concentrations, these two OTUs were more abundant than other transplanted communities. Members of the sulfate-reducing *Desulfobacteraceae* are known to degrade organic matter and release nutrients such as nitrogen and phosphorous (*Pallud & Van Cappellen, 2006*; *Almstrand et al., 2016*). It is possible that the transplantation to a site with lower nitrogen may provide additional metabolic niche space for this organism to increase in abundance or it may simply be a significantly more hospitable environment than its origin. Relatively little is known about the metabolism of the *Anaerolinaceae* family (*Yamada et al., 2006*), but they may also play a role in the terminal mineralization of organic matter in low-oxygen sites (*Sinkko et al., 2013*). The abundance of this OTU also correlated positively with increased temperature between origin and host sites (Fig. 4B), which may suggest that warmer temperatures could facilitate taking advantage of new environmental niches. It is also important to note that the structure of the sediment cores varied between northern and southern sites, and that unknown factors relating to the sediment structure may limit mixing of transplanted and external microbial communities, thus potentially having an effect on the ability of external organisms to invade, colonize, or thrive in transplanted cores. While factors pertaining specifically to the structure of the cores were not measured in this study, we can infer from observed visual differences and from belowground biomass data collected in other studies (*Angermeyer, Crosby & Huber, 2015*; *Crosby et al., 2015*) that northern sites have more un-degraded organic matter than in the south and are likely more porous and oxygen permeable. These differences may play a role in the abundance profiles observed for the *Anaerolinaceae* and *Desulfobacteraceae* OTUs by providing a more anoxic environment when transplanted to the southern sites.

The linear regression results raise the question of why organisms that are strongly correlated to an environmental variable maintain their original relative abundances after transplantation to a site with a large shift in that specific environmental variable. For example, if *Vibrio* does much better in warmer locations, why doesn't it flourish in transplants from the north moved to the south? Our observations highlight two important points. First, it is clear that the established microbial communities in our sediment transplants were resistant to significant invasion from the surrounding host marsh over the time scale of our experiment, especially among the more abundant taxa. However, an ordination analysis of the low abundance OTUs (<5% across all samples) showed there is less definition between the northern and southern sites when these organisms are also considered (Fig. S4). This result suggests that dispersal out of, or invasion into, the community is happening at a low rate, or possibly, that niche occupation is preventing large influxes of diverse colonizing microorganisms (*Brockhurst et al., 2007*). Second, our results suggest that salt marsh communities may be resistant to significant environmental shifts over the course of one growing season. We are beginning to learn more about the temporal dynamics of salt march microbial

communities (*Bowen et al., 2009*), but it is difficult to speculate over what time scales one would expect a transplanted community to become indistinguishable from its host marsh. However, as mentioned above, several studies from a variety of environments have shown that some transplanted communities may not undergo substantial changes, even over relatively long periods of time. However, neither the apparent resistance to invasion nor environmental change rules out the possibility that important, possibly large-scale, shifts are occurring in the active vs. dormant partitions of these microbial communities. It has been shown recently that the activity of salt marsh-associated microbial communities can respond drastically to environmental perturbations (*Kearns et al., 2016*), and this should be taken into account for future transplant studies if our findings of community compositional resilience are reflective of actual dispersal events.

Additional factors that we did not directly test are related to the impact of *Spartina* itself on the associated microbial community, and include the effects of the mortality of the northern transplants and the differences in plant genotypes between regions. The *S. alterniflora* contained in the transplant cores moved from the northern-most two sites to the southern-most two generally died or fared much more poorly than transplants from south to north (*Crosby et al., 2017*). Although we specifically avoided larger *S. alterniflora* roots, our sediment samples likely contained very fine roots that theoretically classify these communities as part of the rhizosphere, and therefore potentially subject to plant-related influences. It is likely that the death of the stems would have an effect on the root uptake of nutrients from the soil and/or the production of exudates, both of which might have an effect on root-associated microbes. We do not know how symbiotically-connected these organisms are in our system, but future transplant experimental designs may be better able to control for plant survival.

Secondly, a recent study by *Bowen et al. (2017)* found that natural microbial communities associated with the rhizosphere of another salt marsh plant, *Phragmites australis*, were strongly regulated by the specific lineage of *P. australis* from which they were sampled. Furthermore, the authors showed that when sterilized plants were grown in a common-garden experiment, they would, over time, accumulate distinct microbial communities that were again defined by the plant lineage. While this work is convincing, it is unclear how applicable the findings are to our *S. alterniflora*-associated microbial communities since *S. alterniflora* along the US East Coast is not divided into multiple obvious lineages. However, there is some evidence that genetic divergence exists in this species between northern and southern populations as demonstrated by haplotype variation (*Blum et al., 2007*), although the exact boundaries between haplotype populations are unclear as are the total extent of genetic differences. Regardless, future studies of microbial ecology in the *S. alterniflora* rhizosphere may greatly benefit from considering plant genotype among the potential environmental factors contributing to the determination of microbial communities.

## CONCLUSION

While there is much work yet to be done in understanding the complex roles that dispersal and environmental selection play in driving the biogeography of

microorganisms (*Evans, Martiny & Allison, 2017*), this study adds new information from an estuarine system to the growing body of evidence that transplanted microbial communities can be resistant to large-scale compositional shifts over seasonal time scales. However, this study also demonstrates that despite the fact that the overall community composition was maintained, the abundances of some potential indicator organisms were strongly influenced by the environmental shifts experienced after transplantation. This tension between general resistance to change (*Bowen et al., 2009*) and the specific adaption of individual species should be a key area of future study in microbial biogeography. It underscores not only the complexity of the big picture interactions between dispersal and environmental selection, but also forces us to consider the cell-level mechanisms that act on dispersed populations to control invasion, colonization and survival in new environments.

## ACKNOWLEDGEMENTS

We thank Rika Anderson, Zoe Cardon, Caroline Fortunato, Heather Leslie, David Rand, Emily Reddington, Sheri Simmons, Joe Vallino, Joe Vineis, and Daniel Weinreich for both laboratory support and guidance during data analysis and interpretation.

### Funding

This research was conducted in the National Estuarine Research Reserve System under an award from the Estuarine Reserves Division, Office of Ocean and Coastal Resource Management, National Ocean Service, and National Oceanic and Atmospheric Administration. Support was also provided through funding to Julie Huber from a Brown-MBL Partnership SEED award, the Neal Cornell Endowed Research Fund, and the NSF Center for Dark Energy Biosphere Investigations (C-DEBI) (OCE-0939564). Additional funding was provided to Sarah Corman-Crosby by the National Park Service George Melendez Wright Climate Change Fellowship. This is C-DEBI Contribution number 426. There was no additional external funding received for this study. The funders had no role in study design, data collection and analysis, decision to publish, or preparation of the manuscript.

### Grant Disclosures

The following grant information was disclosed by the authors:
Neal Cornell Endowed Research Fund.
NSF Center for Dark Energy Biosphere Investigations (C-DEBI): OCE-0939564.
Sarah Corman-Crosby by the National Park Service George Melendez Wright Climate Change Fellowship.

### Competing Interests

Sarah C. Crosby is an employee of Harbor Watch, Earthplace Inc.

## Author Contributions

- Angus Angermeyer conceived and designed the experiments, performed the experiments, analyzed the data, contributed reagents/materials/analysis tools, prepared figures and/or tables, authored or reviewed drafts of the paper, approved the final draft.
- Sarah C. Crosby conceived and designed the experiments, performed the experiments, authored or reviewed drafts of the paper, approved the final draft.
- Julie A. Huber conceived and designed the experiments, contributed reagents/materials/analysis tools, authored or reviewed drafts of the paper, approved the final draft.

## DNA Deposition

The following information was supplied regarding the deposition of DNA sequences:

16S rRNA sequences are publicly available at the "The Visualization and Analysis of Microbial Population Structures" (VAMPS) website (https://vamps.mbl.edu/) (*Huse et al., 2014*) under project title: "JAH_TRP_Bv4v5" and are deposited at the NCBI SRA under accession number PRJNA384656.

## Data Availability

The raw data are provided in the Supplemental Files.

## Supplemental Information

Supplemental information for this article can be found online at http://dx.doi.org/10.7717/peerj.4735#supplemental-information.

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
