# Peer review of "Salt marsh sediment bacterial communities maintain original population structure after transplantation across a latitudinal gradient"

_PeerJ, doi:10.7717/peerj.4735_

## Round 0.1 · original submission · Minor Revisions

The reviewers have made multiple helpful comments and I agree with the majority of them. In particular, I do agree that temperature is one of several variables that may vary along the latitudinal gradient, and so additional context is needed when presenting the results. I also find some of the methodological suggestions to be helpful - although in my opinion it is not necessary to make all of the requested changes, nor conduct all of the suggested analyses. Ultimately this is an interesting and very well-designed study study and I look forward to seeing a revised version of the manuscript.

·

Basic reporting

This manuscript is clear, concise, and well-written throughout. The authors provide the appropriate context of the work and reference the relevant literature. They background sets the stage for their work and provides a rationale for why the work is important. Their hypotheses are clearly articulated and testable. The methods are reasonable and their data analysis seems appropriate for the questions they are asking.

Just a couple minor points:

It might be nice to include in your methods or results, some description about the plant side of the reciprocal transplant. Although you do mention it in the discussion, I did find myself wondering as I was reading your results whether the plants survived the transplant or if there were inhibited from being successful in their new system. It seems from the results that the northern marshes did poorly in the southern system but the southern marshes did fine up north? It might be worth bringing that information in earlier and adding a plant-soil feedback context.

Line 407 - liner should be linear.
Fig. 1 legend - first, figure 1 is actually not referenced in the text. Second, the legend is written essentially as a statement of the main conclusion of the findings, though I assume that the map will be introduced earlier - I looked at the legend and assumed that the figure was going to indicate the effect of transplantation when really it is only a map of sites. Perhaps make the legend a bit more congruent with the content of the figure.

Fig. 3 - I found the P really hard to see in some of these - could the authors use colors instead, to differentiate? It also might be useful to have the northern marshes be open symbols and the southern marshes be closed symbols to make that contrast more clear.

Experimental design

The experimental design for this experiment was straightforward, well described, and clear. All data analysis seems appropriate. I liked the matrix approach.

The authors took sediment cores from five marshes and did a full reciprocal transplant, with all proper controls. They left the transplants out for a growing season and then harvested them to see if environmental factors or dispersal play a dominant role in structuring microbial communities.

Validity of the findings

The results are interesting. Despite a huge latitudinal gradient, the microbial community of transplanted plants retained the signal of there native lands after six months. I wish there were three more years of data to see how long this pattern persisted! I do think a little bit of context for how the plants did would be useful for understanding the broader picture.

Conclusions are not overstated and are well supported by the data.

·

Basic reporting

The article is well-written, clear, and fully cited throughout. The figures look fine, though some larger text could help in some legends. The MS is self-contained as well.

Experimental design

The experimental design meets PeeJ's scopes/aims. The questions are well defined, relevant, and meaningful. The research does address a knowledge gap in salt marsh microbial ecology.

Validity of the findings

The findings are valid, though I think more discussion of the nuances of eh experimental setup is needed. The stats and data are robust and the experiment is well replicated.

I think there's a bit too much speculation in the discussion and could be grounded more in the ltierature.

Additional comments

The study by Angermeyer and colleagues surveys the response of salt marsh sediment bacterial
communities to a large reciprocal transplant experiment between several marshes. They find that the
communities are resistant to transplantation which may suggest limited dispersal of the communities.
This study is well-written, interesting, and the results are generally clear but do need some clarification
at spots. Below I list some major comments and then below I detail some more specific comments
relevant to the text.

Major comment: what ecotype of Spartina was selected for this study? Was it short or tall? This
ultimately has important implications for their study and the interpretation of the data due to the
starkly different habitats that the to ecotypes reside in. If this is the tall ecotype of Spartina then the
plants are experiencing tidal flooding every day, whereas the short ecotype only sees flooding 1/3 of the
month (e.g. Johnson et al. 2016, Ecol Appl).
Given this, if tall was selected the authors are likely seeing full integration of the transplanted soil into
the local conditions of the new marsh. If short was selected, then I do not believe the transplanted soil
fully experienced the abiotic conditions of the marsh itself (e.g. Balser and Firestone 2005, Biogeochem)
and the small change in temperature between sites (17-24C) likely not enough of a push to drive
changes. If the short ecotype was selected the authors would need to demonstrate the exposure of the
transplanted sediments to the new conditions with some measurements of the soil abiotic conditions to
convincingly demonstrate the efficacy of the transplant.
Related to this, I wonder how much exposure to new conditions sediment 20cm deep in marsh sediment
receives. Unless these are bottom up flooding marshes (like many in New England are), these sediments
likely wouldn’t see that drastic of change in their abiotic conditions in particular because the biggest
deciding factor here (air temperature) likely isn’t all that differ 20cm deep.

Title: temperature is just one component of what makes up the gradient sampled here, especially in
terms of the soil chemistry. I would suggest revising the title to reflect this. Also, population is the wrong
word here, community is more appropriate. Something like “Salt marsh bacteria communities maintain
structure despite transplantation’.

Methods
General comment: the authors should probably remove Chloroplasts from this analysis. V4 is generally
terrible for this group of taxa and the short-read length makes meaningful identification of these
eukaryotes difficult at best. Also, why mantel tests in lieu of something like a
PERMANOVA/PERMDISP/ANOSIM?
Line 139: did this include a plant(s) or was it just bare sediment?
Line 140: it would be helpful to know a bit more about where in the marsh they were transplanted.
Things like height relative mean sea level, tidal inundation, distance from the creek bank, and a bit more
about the marshes themselves to better interpret the experimental design.
Line 151: I assume this is just something like cheese cloth, but it would be helpful to know the mesh size
here.Line 155: So the authors bored 20cm into the sediment with a spatula? How are they sure they reached
the desired depth and didn’t get mixing of higher layers? The sampling says 20cm, but was it around
20cm (e.g. 19-21cm)?
Line 178: why the DNA/RNA kit when only DNA is presented?
Line 188: delete ‘Benchtop’.
Line 190: I would find it helpful for the authors to just include what they did instead of having to go to
GitHub/another paper.
Line 192: data was rarefied before OTU picking? This seems odd and potentially biasing OTU clustering
by removal of diversity. Can the authors please clarify?
Line 193: OTU picking method? Closed pick? De novo pick?
Line 222: this is on rarefied data, no? Absolute abundances in this instance seem to be perfectly viable
over relativized abundance.
Line 237: It would also seem more appropriate to use a ANCOVA for the environmental variables and
OTUs, holding origin and site as effects. Given that the variables change as a function of site and covary,
this should be taken into account in the regression model.
Line 241: Bonferroni corrections are notoriously overly conservative, did the authors try another p-adjustment technique like Benjamini-Hochberg?

Results
Line 250: remove ‘host’
Line 251: I also assume no roots entered the cores as well?
Line 258: Was anoxia measured? Creek bank sediment is only usually anoxic during flooding.
Line 275-276: is this information about DNA mass needed?
Line 283: this paragraph could be strengthened with some statistics (e.g. Kruskal-Wallis test w/pcorrection).
Line 284: Drop the ‘s’ from the end of ‘compositions’.
Line 287: not a requirement, but oligotyping would be interesting for this Vibrio OTU.
Line 314 (and in methods too): is there a reason the ordination would need to be done with all OTUs and
then again with the top ten most abundant taxa? Bray-Curtis is most strongly influenced by the top taxa
and given that the ordinations look very similar, I question whether it is needed because the authors do
not do anything with these top ten taxa.
Line 321: indicator OTUs is misleading here. This implies that they are indicator species, as determined
by that specific test. I don’t disagree with how the authors determined these four OTUs, however I’m
surprised that only 4 of the 21k OTUs were significantly correlated.

Discussion
Line 336: this paragraph rehashes the introduction.
Line 345: one thing that’s hard to tell about this experiment is whether dispersal, due to tidal flooding,
occurred especially at the depth sampled. Do the authors have data about the sediment chemistry
outside
Line 346: this is not just a temperature gradient; many things change along the eastern US coast.
Line 350-352: without presenting more data about the sites (e.g. porewater chemistry, ocean water
chemistry, variations in abiotic conditions, ect), this is a hard claim to make. The physical properties of
the sediments themselves, as noted in the methods/results, suggest these sites are very different.
Line 355: this implies that sampling didn’t disrupt the communities, not that the methods were effective
at excambing the question. Which ultimately is a strength given the likely release of carbon via severing
marsh roots.
Line 362: while this intuitively makes sense, the authors lack the resolution in terms of measured
variables to really parse out the driving factors here.
Line 365: remove ‘host’.
Lines 368-370: this sentence is confusing, I’d rephrase or remove it because the next sentence says all
that’s needed.
Line 378: one thing about this paragraph that the authors do not discuss is the idea of no mixing of the
local conditions. This has been discussed by several papers (e.g. Balser and Firestone 2005, Soil Biochem;
Hughes and Martiny 2013, ISME J; Gasol et al. 2002, Limnol Oceano). Given this, it is likely that their
results show no mixing. In addition, a recent paper about dispersal in soil systems would also be
appropriate for discussion (Evans et al 2017, ISME J).
Line 402: while Vibrio do bloom in the summer, the authors sampled in October. Work by Bowen et al
2009, ISMEJ showed seasonal patterns in the microbial communities in marshes, which was also show in
the active communities by Kearns et al. 2016, Nat Commun. Given this, how confident can we be that at
the end of the season samplings are reflective of the warmer temperatures?
Further, given the lack of change in transplanted cores it suggests its likely something about the soil
chemistry instead of the temperature. This may be reflective of the sandy nature of southern marshes
relative to the high carbon marshes in the north.
Line 438: the presence of Vibrio regardless of temperature suggests lack of mixing, which may be why
the persist. I also think that the idea of resistance, which has been shown a few times for marsh
communities (Bowen et al. 2009, 2011, ISME; Kearns et al. 2016, Nat Commun; Angell et al. 2018, Front
Micro), is an interesting talking point here.
Line 458: while I find this paragraph interesting, I question how applicable it is to this study given that
the majority of studies in this area of plant-microbe interactions, outside of the work of Bowen and
colleagues, shows the environment overrides plant genetic effects every time. Further, based on the sampling, it appears this study would be independent of the plant itself since the authors do not
mention roots and the rhizosphere effect only extending a few millimeters from the root surface.
One other point about this is that Phragmites is hardly a salt marsh plant as its home range is in brackish
(<10ppt) to freshwater environments, which all Spartina would be unable to compete in.

Figures
Fig 1: looks good. Perhaps scale bar?
Fig 2: I would recommend removing the failed samples, its not needed and it distracts from the figure.
Also the size of the text could be improved.
Fig 3: the shape/letter makes these hard to interpret at first site. Perhaps color them?
Fig 4: I would add a legend, it’s a pint to look back and forth between this figure and figure 2.
Table 1: are these averages over the course of the experiment or just a one time point measurement? If
an average, please include the SE or SD.

---

## Round 0.2 · accepted · Accept

The authors have successfully addressed the reviewers' comments, and I look forward to seeing this published

#